# LEARNING DEEP RESNET BLOCKS SEQUENTIALLY USING BOOSTING THEORY

## ABSTRACT

We prove a multiclass boosting theory for the ResNet architectures which simultaneously creates a new technique for multiclass boosting and provides a new algorithm for ResNet-style architectures. Our proposed training algorithm, *BoostResNet*, is particularly suitable in non-differentiable architectures. Our method only requires the relatively inexpensive sequential training of $T$ "shallow ResNets". We prove that the training error decays exponentially with the depth $T$ if the weak module classifiers that we train perform slightly better than some weak baseline. In other words, we propose a weak learning condition and prove a boosting theory for ResNet under the weak learning condition. A generalization error bound based on margin theory is proved and suggests that ResNet could be resistant to overfitting using a network with $l_1$ norm bounded weights.

## 1 INTRODUCTION

Why do residual neural networks (ResNets) (He et al., 2016) and the related highway networks (Srivastava et al., 2015) work? And if we study closely why they work, can we come up with new understandings of how to train them and how to define working algorithms?

Deep neural networks have elicited breakthrough successes in machine learning, especially in image classification and object recognition (Krizhevsky et al., 2012; Sermanet et al., 2013; Simonyan & Zisserman, 2014; Zeiler & Fergus, 2014) in recent years. As the number of layers increases, the nonlinear network becomes more powerful, deriving richer features from input data. Empirical studies suggest that challenging tasks in image classification (He et al., 2015; Ioffe & Szegedy, 2015; Simonyan & Zisserman, 2014; Szegedy et al., 2015) and object recognition (Girshick, 2015; Girshick et al., 2014; He et al., 2014; Long et al., 2015; Ren et al., 2015) often require "deep" networks, consisting of tens or hundreds of layers. Theoretical analyses have further justified the power of deep networks (Mhaskar & Poggio, 2016) compared to shallow networks.

However, deep neural networks are difficult to train despite their intrinsic representational power. Stochastic gradient descent with back-propagation (BP) (LeCun et al., 1989) and its variants are commonly used to solve the non-convex optimization problems. A major challenge that exists for training both shallow and deep networks is *vanishing* or *exploding gradients* (Bengio et al., 1994; Glorot & Bengio, 2010). Recent works have proposed normalization techniques (Glorot & Bengio, 2010; LeCun et al., 2012; Ioffe & Szegedy, 2015; Saxe et al., 2013) to effectively ease the problem and achieve convergence. In training deep networks, however, a surprising *training performance degradation* is observed (He & Sun, 2015; Srivastava et al., 2015; He et al., 2016): the training performance degrades rapidly with increased network depth after some saturation point. This training performance degradation is representationally surprising as one can easily construct a deep network identical to a shallow network by forcing any part of the deep network to be the same as the shallow network with the remaining layers functioning as identity maps. He et al. (He et al., 2016) presented a *residual network* (ResNet) learning framework to ease the training of networks that are substantially deeper than those used previously. And they explicitly reformulate the layers as learning residual functions with reference to the layer inputs by adding identity loops to the layers. It is shown in (Hardt & Ma, 2016) that identity loops ease the problem of spurious local optima in shallow networks. Srivastava et al. (Srivastava et al., 2015) introduce a novel architecture that enables the optimization of networks with virtually arbitrary depth through the use of a learned gating mechanism for regulating information flow.

Empirical evidence overwhelmingly shows that these deep residual networks are easier to optimize than non-residual ones. Can we develop a theoretical justification for this observation? And does that justification point us towards new algorithms with better characteristics?

## 1.1 SUMMARY OF RESULTS

We propose a new framework, *multi-channel telescoping sum boosting* (defined in Section 4), to characterize a feed forward ResNet in Section 3. We show that the top level (final) output of a ResNet can be thought of as a layer-by-layer boosting method (defined in Section 2). Error bounds for telescoping sum boosting are provided.

We introduce a learning algorithm (*BoostResNet*) guaranteed to reduce error exponentially as depth increases so long as a weak learning assumption is obeyed. BoostResNet adaptively selects training samples or changes the cost function (Section 4 Theorem 4.2). In Section 4.4, we analyze the generalization error of BoostResNet and provide advice to avoid overfitting. The procedure trains each residual block sequentially, only requiring that each provides a better-than-a-weak-baseline in predicting labels.

BoostResNet requires radically lower computational complexity for training than end-to-end back propagation (*e2eBP*). Memorywise, *BoostResNet* requires only individual layers of the network to be in the graphics processing unit (GPU) while *e2eBP* inevitably keeps all layers in the GPU. For example, in a state-of-the-art deep ResNet, this might reduce the RAM requirements for GPU by a factor of the depth of the network. Similar improvements in computation are observed since each *e2eBP* step involves back propagating through the entire deep network.

Experimentally, we compare *BoostResNet* with *e2eBP* over two types of feed-forward ResNets, *multilayer perceptron residual network* (MLP-ResNet) and *convolutional neural network residual network* (CNN-ResNet), on multiple datasets. *BoostResNet* shows substantial computational performance improvements and accuracy improvement under the MLP-ResNet architecture. Under *CNN-ResNet*, a faster convergence for *BoostResNet* is observed.

One of the hallmarks of our approach is to make an explicit distinction between the classes of the multiclass learning problem and *channels* that are constructed by the learning procedure. A channel here is essentially a scalar value modified by the rounds of boosting so as to implicitly minimize the multiclass error rate. Our *multi-channel telescoping sum boosting* learning framework is not limited to ResNet and can be extended to other, even non-differentiable, nonlinear hypothesis units, such as decision trees or tensor decompositions.

## 1.2 RELATED WORKS

Training deep neural networks has been an active research area in the past few years. The main optimization challenge lies in the highly non-convex nature of the loss function. There are two main ways to address this optimization problem: one is to select a loss function and network architecture that have better geometric properties (details refer to appendix A.1), and the other is to improve the network's learning procedure (details refer to appendix A.2).

Many authors have previously looked into neural networks and boosting, each in a different way. Bengio et al. (2006) introduce single hidden layer convex neural networks, and propose a gradient boosting algorithm to learn the weights of the linear classifier. The approach has not been generalized to deep networks with more than one hidden layer. Shalev-Shwartz (2014) proposes a selfieBoost algorithm which boosts the accuracy of an entire network. Our algorithm is different as we instead construct ensembles of classifiers. Veit et al. (2016) interpret residual networks as a collection of many paths of differing length. Their empirical study shows that residual networks avoid the vanishing gradient problem by introducing short paths which can carry gradient throughout the extent of very deep networks. The authors of AdaNet (Cortes et al., 2016) consider ensembles of neural layers with a boosting-style algorithm and provide a method for structural learning of neural networks by optimizing over the generalization bound, which consists of the training error and the complexity of the AdaNet architecture. AdaNet uses the traditional boosting framework where weak classifiers are being boosted. Therefore, to obtain low training error guarantee, AdaNet maps the feature vectors (hidden layer representations) to a classifier space and boosts the weak classifiers. Our BoostResNet, instead, boosts representations (feature vectors) over multiple channels, and therefore produces a less "bushy" architecture. BoostResNet focuses on a ResNet architecture, provides a new training algorithm for ResNet, and proves a training error guarantee for deep ResNet architecture. A ResNet-style architecture is a special case of AdaNet, so AdaNet generalization guarantee applies here and our generalization analysis is built upon their work.

## 2 PRELIMINARIES

A *residual neural network* (ResNet) is composed of stacked entities referred to as residual blocks. Each residual block consists of a neural network module and an identity loop (shortcut). Commonly used modules include MLP and CNN. Throughout this paper, we consider training and test examples generated i.i.d. from some distribution $\mathcal{D}$ over $\mathcal{X} \times \mathcal{Y}$, where $\mathcal{X}$ is the input space and $\mathcal{Y}$ is the label space. We denote by $S = ((x_1, y_1), (x_2, y_2), \ldots, (x_m, y_m))$ a training set of $m$ examples drawn according to $\mathcal{D}^m$.

**A Residual Block of ResNet**    ResNet consists of residual blocks. Each residual block contains a module and an identity loop. Let each module map its input $\tilde{x}$ to $f_t(\tilde{x})$ where $t$ denotes the level of the modules. Each module $f_t$ is a nonlinear unit with $n$ channels, i.e., $f_t(\cdot) \in \mathbb{R}^n$. In *multilayer perceptron residual network* (MLP-ResNet), $f_t$ is a shallow MLP, for instance, $f_t(\tilde{x}) = \tilde{V}_t^\top \sigma(\tilde{W}_t^\top \tilde{x})$ where $\tilde{W}_t \in \mathbb{R}^{n \times k}$, $\tilde{V}_t \in \mathbb{R}^{k \times n}$ and $\sigma$ is a nonlinear operator such as sigmoidal function or relu function. Similarly, in *convolutional neural network residual network* (CNN-ResNet), $f_t(\cdot)$ represents the $t$-th convolutional module. Then the $t$-th residual block outputs $g_{t+1}(x)$

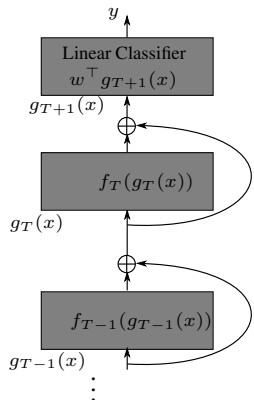

**Figure 1:** The architecture of a residual network (ResNet).

$$g_{t+1}(x) = f_t(g_t(x)) + g_t(x), \tag{1}$$

where $x$ is the input fed to the ResNet. See Figure 1 for an illustration of a ResNet, which consists of stacked residual blocks (each residual block contains a nonlinear module and an identity loop).

**Output of ResNet**    Due to the recursive relation specified in Equation (1), the output of the $T$-th residual block is equal to the summation over lower module outputs, i.e., $g_{T+1}(x) = f_T(g_T(x)) + g_T(x) = \sum_{t=1}^{T} f_t(g_t(x))$, where $g_1(x) = x$. For binary classification tasks, the final output of a ResNet given input $x$ is rendered after a linear classifier $\mathbf{w} \in \mathbb{R}^n$ on representation $g_{T+1}(x)$ (In the multiclass setting, let $C$ be the number of classes; the linear classifier $W \in \mathbb{R}^{n \times C}$ is a matrix instead of a vector.):

$$\widehat{y} = \tilde{\sigma}(F(x)) = \tilde{\sigma}(\mathbf{w}^\top g_{T+1}(x)) = \tilde{\sigma}\left(\mathbf{w}^\top \sum_{t=1}^{T} f_t(g_t(x))\right) \tag{2}$$

where $F(x) = \mathbf{w}^\top g_{T+1}(x)$ and $\tilde{\sigma}(\cdot)$ denotes a map from classifier outputs (scores) to labels. For instance $\tilde{\sigma}(z) = \text{sign}(z)$ for binary classification ($\tilde{\sigma}(z) = \arg\max_i z_i$ for multiclass classification). The parameters of a depth-$T$ ResNet are $\{\mathbf{w}, \{f_t(\cdot), \forall t \in T\}\}$. A ResNet training involves training the classifier $\mathbf{w}$ and the weights of modules $f_t(\cdot) \, \forall t \in [T]$ when training examples $(x_1, y_1), (x_2, y_2), \ldots, (x_m, y_m)$ are available.

**Boosting**    Boosting (Freund & Schapire, 1995) assumes the availability of *a weak learning algorithm* which, given labeled training examples, produces a *weak classifier* (a.k.a. *base classifier*). The goal of boosting is to improve the performance of the weak learning algorithm. The key idea behind boosting is to choose training sets for the weak classifier in such a fashion as to force it to infer something new about the data each time it is called. The weak learning algorithm will finally combine many weak classifiers into a single *strong classifier* whose prediction power is strong.

From empirical experience, ResNet remedies the problem of training error degradation (instability of solving non-convex optimization problem using SGD) in deeper neural networks. We are curious about whether there is a theoretical justification that identity loops help in training. More importantly, we are interested in proposing a new algorithm that avoids end-to-end back-propagation (*e2eBP*) through the deep network and thus is immune to the instability of SGD for non-convex optimization of deep neural networks.

## 3 RESNET IN TELESCOPING SUM BOOSTING FRAMEWORK

As we recall from Equation (2), ResNet indeed has a similar form as the strong classifier in boosting. The key difference is that boosting is an ensemble of estimated hypotheses whereas ResNet is an

ensemble of estimated feature representations $\sum_{t=1}^{T} f_t(g_t(x))$. To solve this problem, we introduce an auxiliary linear classifier $\mathbf{w}_t$ on top of each residual block to construct a *hypothesis module*. Formally, a *hypothesis module* is defined as

$$o_t(x) \stackrel{\text{def}}{=} \mathbf{w}_t^\top g_t(x) \in \mathbb{R} \tag{3}$$

in the binary classification setting. Therefore $o_{t+1}(x) = \mathbf{w}_{t+1}^\top[f_t(g_t(x)) + g_t(x)]$ as $g_{t+1}(x) = f_t(g_t(x)) + g_t(x)$. We emphasize that given $g_t(x)$, we only need to train $f_t$ and $\mathbf{w}_{t+1}$ to train $o_{t+1}(x)$. In other words, we feed the output of previous residual block ($g_t(x)$) to the current module and train the weights of current module $f_t(\cdot)$ and the auxiliary classifier $\mathbf{w}_{t+1}$.

Now the input, $g_{t+1}(x)$, of the $t + 1$-th residual block is the output, $f_t(g_t(x)) + g_t(x)$, of the $t$-th residual block. As a result, $o_t(x) = \sum_{t'=1}^{t-1} \mathbf{w}_t^\top f_{t'}(g_{t'}(x))$. In other words, the auxiliary linear classifier is common for all modules underneath. It would not be realistic to assume a common auxiliary linear classifier, as such an assumption prevents us from training the $T$ hypothesis module sequentially. We design a ***weak module classifier*** using the idea of telescoping sum as follows.

**Definition 3.1.** *A weak module classifier is defined as*

$$h_t(x) \stackrel{\text{def}}{=} \alpha_{t+1}o_{t+1}(x) - \alpha_t o_t(x) \tag{4}$$

*where $o_t(x) \stackrel{\text{def}}{=} \mathbf{w}_t^\top g_t(x)$ is a hypothesis module, and $\alpha_t$ is a scalar. We call it a "telescoping sum boosting" framework if the weak learners are restricted to the form of the weak module classifier.*

**ResNet: Ensemble of Weak Module Classifiers**   Recall that the $T$-th residual block of a ResNet outputs $g_{T+1}(x)$, which is fed to the top/final linear classifier for the final classification. We show that an ensemble of the weak module classifiers is equivalent to a ResNet's final output. We state it formally in Lemma 3.2. For purposes of exposition, we will call $F(x)$ the output of ResNet although a $\tilde{\sigma}$ function is applied on top of $F(x)$, mapping the output to the label space $\mathcal{Y}$.

**Lemma 3.2.** *Let the input $g_t(x)$ of the $t$-th module be the output of the previous module, i.e., $g_{t+1}(x) = f_t(g_t(x)) + g_t(x)$. Then the summation of $T$ weak module classifiers divided by $\alpha_{T+1}$ is identical to the output, $F(x)$, of the depth-$T$ ResNet,*

$$F(x) = \mathbf{w}^\top g_{T+1}(x) \equiv \frac{1}{\alpha_{T+1}} \sum_{t=1}^{T} h_t(x), \tag{5}$$

*where the weak module classifier $h_t(x)$ is defined in Equation (4).*

See Appendix B for the proof. Overall, our proposed ensemble of weak module classifiers is a new framework that allows for sequential training of ResNet. Note that traditional boosting algorithm results do not apply here. We now analyze our telescoping sum boosting framework in Section 4. Our analysis applies to both binary and multiclass, but we will focus on the binary class for simplicity in the main text and defer the multiclass analysis to the Appendix F.

## 4 TELESCOPING SUM BOOSTING FOR BINARY CLASSIFICATION

Below, we propose a learning algorithm whose training error decays exponentially with the number of weak module classifiers $T$ under a weak learning condition. We restrict to bounded hypothesis modules, i.e., $|o_t(x)| \leq 1$.

### 4.1 WEAK LEARNING CONDITION

The weak module classifier involves the difference between (scaled version of) $o_{t+1}(x)$ and $o_t(x)$. Let $\tilde{\gamma}_t \stackrel{\text{def}}{=} \mathbb{E}_{i \sim D_{t-1}}[y_i o_t(x_i)] > 0$ be the *edge* of the hypothesis module $o_t(x)$, where $D_{t-1}$ is the weight of the examples. As the hypothesis module $o_t(x)$ is bounded by 1, we obtain $|\tilde{\gamma}_t| \leq 1$. So $\tilde{\gamma}_t$ characterizes the performance of the hypothesis module $o_t(x)$. A natural requirement would be that $o_{t+1}(x)$ improves slightly upon $o_t(x)$, and thus $\tilde{\gamma}_{t+1} - \tilde{\gamma}_t \geq \gamma' > 0$ could serve as a weak learning condition. However this weak learning condition is too strong: even when current hypothesis module is performing almost ideally ($\tilde{\gamma}_t$ is close to 1), we still seek a hypothesis module which performs consistently better than the previous one by $\gamma'$. Instead, we consider a much weaker learning condition, inspired by training error analysis, as follows.

**Definition 4.1** ($\gamma$-Weak Learning Condition). *A weak module classifier $h_t(x) = \alpha_{t+1}o_{t+1} - \alpha_t o_t$ satisfies the $\gamma$-weak learning condition if $\frac{\tilde{\gamma}_{t+1}^2 - \tilde{\gamma}_t^2}{1 - \tilde{\gamma}_t^2} \geq \gamma^2 > 0$ and the covariance between $\exp(-yo_{t+1}(x))$ and $\exp(yo_t(x))$ is non-positive.*

The weak learning condition is motivated by the learning theory and it is met in practice (refer to Figure 4).

**Interpretation of weak learning condition** For each weak module classifier $h_t(x)$, $\gamma_t \overset{\text{def}}{=} \sqrt{\frac{\tilde{\gamma}_{t+1}^2 - \tilde{\gamma}_t^2}{1 - \tilde{\gamma}_t^2}}$ characterizes the normalized improvement of the correlation between the true labels $y$ and the hypothesis modules $o_{t+1}(x)$ over the correlation between the true labels $y$ and the hypothesis modules $o_t(x)$. The condition specified in Definition 4.1 is mild as it requires the hypothesis module $o_{t+1}(x)$ to perform only slightly better than the previous hypothesis module $o_t(x)$. In residual network, since $o_{t+1}(x)$ represents a depth-$(t+1)$ residual network which is a deeper counterpart of the depth-$t$ residual network $o_t(x)$, it is natural to assume that the deeper residual network improves slightly upon the shallower residual network. When $\tilde{\gamma}_t$ is close to 1, $\tilde{\gamma}_{t+1}^2$ only needs to be slightly better than $\tilde{\gamma}_t^2$ as the denominator $1 - \tilde{\gamma}_t^2$ is small. The assumption of the covariance between $\exp(-yo_{t+1}(x))$ and $\exp(yo_t(x))$ being non-positive is suggesting that the weak module classifiers should not be adversarial, which may be a reasonable assumption for ResNet.

### 4.2 BOOSTRESNET

We now propose a novel training algorithm for telescoping sum boosting under binary-class classification as in Algorithm 1. In particular, we introduce a training procedure for deep ResNet in Algorithm 1 & 2, **BoostResNet**, which only requires sequential training of shallow ResNets.

The training algorithm is a *module-by-module* procedure following a *bottom-up* fashion as the outputs of the $t$-th module $g_{t+1}(x)$ are fed as the training examples to the next $t+1$-th module. Each of the shallow ResNet $f_t(g_t(x)) + g_t(x)$ is combined with an auxiliary linear classifier $\mathbf{w}_{t+1}$ to form a hypothesis module $o_{t+1}(x)$. The weights of the ResNet are trained on these shallow ResNets. The telescoping sum construction is the key for successful interpretation of ResNet as ensembles of weak module classifiers. The innovative introduction of the auxiliary linear classifiers ($\mathbf{w}_{t+1}$) is the key solution for successful multi-channel representation boosting with theoretical guarantees. Auxiliary linear classifiers are only used to guide training, and they are not included in the model (proved in Lemma 3.2). This is the fundamental difference between BoostResNet and AdaNet. AdaNet (Cortes et al., 2016) maps the feature vectors (hidden layer representations) to a classifier space and boosts the weak classifiers. Our framework is a multi-channel representation (or information) boosting rather than a traditional classifier boosting. Traditional boosting theory does not apply in our setting.

---

**Algorithm 1** BoostResNet: telescoping sum boosting for binary-class classification

---

**Input:** $m$ labeled samples $[(x_i, y_i)]_m$ where $y_i \in \{-1, +1\}$ and a threshold $\gamma$
**Output:** $\{f_t(\cdot), \forall t\}$ and $\mathbf{w}_{T+1}$                    ▷ Discard $\mathbf{w}_{t+1}, \forall t \neq T$
 1: Initialize $t \leftarrow 0$, $\tilde{\gamma}_0 \leftarrow 0$, $\alpha_0 \leftarrow 0$, $o_0(x) \leftarrow 0$
 2: Initialize sample weights at round 0: $D_0(i) \leftarrow 1/m, \forall i \in [m]$
 3: **while** $\gamma_t > \gamma$ **do**
 4:     $f_t(\cdot), \alpha_{t+1}, \mathbf{w}_{t+1}, o_{t+1}(x) \leftarrow$ Algorithm 2($g_t(x), D_t, o_t(x), \alpha_t$)
 5:     Compute $\gamma_t \leftarrow \sqrt{\frac{\tilde{\gamma}_{t+1}^2 - \tilde{\gamma}_t^2}{1 - \tilde{\gamma}_t^2}}$            ▷ where $\tilde{\gamma}_{t+1} \leftarrow \mathbb{E}_{i \sim D_t}[y_i o_{t+1}(x_i)]$
 6:     Update $D_{t+1}(i) \leftarrow \frac{D_t(i) \exp(-y_i h_t(x_i))}{\sum_{i=1}^{m} D_t(i) \exp[-y_i h_t(x_i)]}$        ▷ where $h_t(x) = \alpha_{t+1}o_{t+1}(x) - \alpha_t o_t(x)$
 7:     $t \leftarrow t + 1$
 8: **end while**
 9: $T \leftarrow t - 1$

---

**Theorem 4.2.** [ *Training error bound* ] *The training error of a $T$-module telescoping sum boosting framework using Algorithms 1 and 2 decays exponentially with the number of modules $T$,*

$$\Pr_{i \sim S}\left(\tilde{\sigma}\left(\sum_t h_t(x_i)\right) \neq y_i\right) \leq e^{-\frac{1}{2}T\gamma^2}$$

---

**Algorithm 2** BoostResNet: oracle implementation for training a ResNet module

---

**Input:** $g_t(x), D_t, o_t(x)$ and $\alpha_t$
**Output:** $f_t(\cdot), \alpha_{t+1}, \mathbf{w}_{t+1}$ and $o_{t+1}(x)$
1: $(f_t, \alpha_{t+1}, \mathbf{w}_{t+1}) \leftarrow \arg \min_{(f, \alpha, \mathbf{v})} \sum_{i=1}^{m} D_t(i) \exp \left( -y_i \alpha \mathbf{v}^\top \left[ f(g_t(x_i)) + g_t(x_i) \right] + y_i \alpha_t o_t(x_i) \right)$
2: $o_{t+1}(x) \leftarrow \mathbf{w}_{t+1}^\top \left[ f_t(g_t(x)) + g_t(x) \right]$

---

*if $\forall t \in [T]$ the weak module classifier $h_t(x)$ satisfies the $\gamma$-weak learning condition defined in Definition 4.1.*

The training error of Algorithms 1 and 2 is guaranteed to decay exponentially with the ResNet depth even when each hypothesis module $o_{t+1}(x)$ performs slightly better than its previous hypothesis module $o_t(x)$ (i.e., $\gamma > 0$). Refer to Appendix F for the algorithm and theoretical guarantees for multiclass classification.

### 4.3 ORACLE IMPLEMENTATION FOR RESNET

In Algorithm 2, the implementation of the oracle at line 1 is equivalent to

$$(f_t, \alpha_{t+1}, \mathbf{w}_{t+1}) = \arg \min_{(f, \alpha, \mathbf{v})} \frac{1}{m} \sum_{i=1}^{m} \exp \left( -y_i \alpha \mathbf{v}^\top \left[ f(g_t(x_i)) + g_t(x_i) \right] \right) \quad (6)$$

The minimization problem over $f$ corresponds to finding the weights of the $t$-th nonlinear module of the residual network. Auxiliary classifier $\mathbf{w}_{t+1}$ is used to help solve this minimization problem with the guidance of training labels $y_i$. However, the final neural network model includes none of the auxiliary classifiers, and still follows a standard ResNet structure (proved in Lemma 3.2). In practice, there are various ways to implement Equation (6). For instance, Janzamin et. al. (Janzamin et al., 2015) propose a tensor decomposition technique which decomposes a tensor formed by some transformation of the features $x$ combined with labels $y$ and recovers the weights of a one-hidden layer neural network with guarantees. One can also use back-propagation as numerous works have shown that gradient based training are relatively stable on shallow networks with identity loops (Hardt & Ma, 2016; He et al., 2016).

**Computational and Memory Efficiency** It is worth noting that *BoostResNet* training is memory efficient as the training process only requires parameters of two consecutive residual blocks to be in memory. Given that the limited GPU memory being one of the main bottlenecks for computational efficiency, *BoostResNet* requires significantly less training time than *e2eBP* in deep networks as a result of reduced communication overhead and the speed-up in shallow gradient forwarding and back-propagation. Let $M_1$ be the memory required for one module, and $M_2$ be the memory required for one linear classifier, the memory consumption is $M_1 + M_2$ by *BoostResNet* and $M_1 T + M_2$ by *e2eBP*. Let the flops needed for gradient update over one module and one linear classifier be $C_1$ and $C_2$ respectively, the computation cost is $C_1 + C_2$ by *BoostResNet* and $C_1 T + C_2$ by *e2eBP*.

### 4.4 GENERALIZATION ERROR ANALYSIS

In this section, we analyze the generalization error to understand the possibility of overfitting under Algorithm 1. The strong classifier or the ResNet is $F(x) = \frac{\sum_t h_t(x)}{\alpha_{T+1}}$. Now we define the *margin* for example $(x, y)$ as $yF(x)$. For simplicity, we consider MLP-ResNet with $n$ multiple channels and assume that the weight vector connecting a neuron at layer $t$ with its preceding layer neurons is $l_1$ norm bounded by $\Lambda_{t,t-1}$. Recall that there exists a linear classifier $\mathbf{w}$ on top, and we restrict to $l_1$ norm bounded classifiers, i.e., $\|\mathbf{w}\|_1 \leq C_0 < \infty$. The expected training examples are $l_\infty$ norm bounded $r_\infty \stackrel{\text{def}}{=} \mathbb{E}_{S \sim \mathcal{D}} \left[ \max_{i \in [m]} \|x_i\|_\infty \right] < \infty$. We introduce Corollary 4.3 which follows directly from Lemma 2 of (Cortes et al., 2016).

**Corollary 4.3.** *(Cortes et al., 2016) Let $\mathcal{D}$ be a distribution over $\mathcal{X} \times \mathcal{Y}$ and $\mathcal{S}$ be a sample of $m$ examples chosen independently at random according to $\mathcal{D}$. With probability at least $1 - \delta$, for $\theta > 0$,*

*the strong classifier $F(x)$ (ResNet) satisfies that*

$$\Pr_{\mathcal{D}}\left(yF(x) \leq 0\right) \leq \Pr_{S}\left(yF(x) \leq \theta\right) + \frac{4C_0 r_\infty}{\theta}\sqrt{\frac{\log(2n)}{2m}}\sum_{t=1}^{T}\Lambda_t + \frac{2}{\theta}\sqrt{\frac{\log T}{m}} + \beta(\theta, m, T, \delta) \quad (7)$$

*where $\Lambda_t \stackrel{def}{=} \prod_{t'=1}^{t} 2\Lambda_{t',t'-1}$ and $\beta(\theta, m, T, \delta) \stackrel{def}{=} \sqrt{\left\lceil \frac{4}{\theta^2}\log\left(\frac{\theta^2 m}{\log T}\right)\right\rceil \frac{\log T}{m} + \frac{\log\frac{2}{\delta}}{2m}}$.*

From Corollary 4.3, we obtain a generalization error bound in terms of margin bound $\Pr_S\left(yF(x) \leq \theta\right)$ and network complexity $\frac{4C_0 r_\infty}{\theta}\sqrt{\frac{\log(2n)}{2m}}\sum_{t=1}^{T}\Lambda_t + \frac{2}{\theta}\sqrt{\frac{\log T}{m}} + \beta(\theta, m, T, \delta)$. Larger margin bound (larger $\theta$) contributes positively to generalization accuracy, and $l_1$ norm bounded weights (smaller $\sum_{t=1}^{T}\Lambda_t$ ) are beneficial to control network complexity and to avoid overfitting. The dominant term in the network complexity is $\frac{4C_0 r_\infty}{\theta}\sqrt{\frac{\log(2n)}{2m}}\sum_{t=1}^{T}\Lambda_t$ which scales as least linearly with the depth $T$. See appendix D for the proof.

This corollary suggests that stronger weak module classifiers which produce higher accuracy predictions and larger edges, will yield larger margins and suffer less from overfitting. The larger the value of $\theta$, the smaller the term $\frac{4C_0 r_\infty}{\theta}\sqrt{\frac{\log(2n)}{2m}}\sum_{t=1}^{T}\Lambda_t + \frac{2}{\theta}\sqrt{\frac{\log T}{m}} + \beta(\theta, m, T, \delta)$ is. With larger edges on the training set and when $\tilde{\gamma}_{T+1} < 1$, we are able to choose larger values of $\theta$ while keeping the error term zero or close to zero.

## 5 EXPERIMENTS

We compare our proposed *BoostResNet* algorithm with *e2eBP* training a ResNet on the MNIST (LeCun et al., 1998), street view house numbers (SVHN) (Netzer et al., 2011), and CIFAR-10 (Krizhevsky & Hinton, 2009) benchmark datasets. Two different types of architectures are tested: a ResNet where each module is a fully-connected multi-layer perceptron (MLP-ResNet) and a more common, convolutional neural network residual network (CNN-ResNet). In each experiment the architecture of both algorithms is identical, and they are both initialized with the same random seed. As a baseline, we also experiment with standard boosting (AdaBoost.MM Mukherjee & Schapire (2013)) of convolutional modules in appendix H.2 for SVHN and CIFAR-10 datasets. Our experiments are programmed in the Torch deep learning framework for Lua and executed on NVIDIA Tesla P100 GPUs. All models are trained using the Adam variant of SGD Kingma & Ba (2014).

**MLP-ResNet on MNIST** The MNIST database (LeCun et al., 1998) of handwritten digits has a training set of 60,000 examples, and a test set of 10,000 examples. The data contains ten classes. We test the performance of *BoostResNet* on MLP-ResNet using MNIST dataset, and compare it with *e2eBP* baseline. Each residual block is composed of an MLP with a single, 1024-dimensional hidden layer. The training and test error between *BoostResNet* and *e2eBP* is in Figure 2 as a function of depth. Surprisingly, we find that training error degrades for *e2eBP*, although the ResNet's identity loop is supposed to alleviate this problem. Our proposed sequential training procedure, *BoostResNet*, relieves gradient instability issues, and continues to perform well as depth increases.

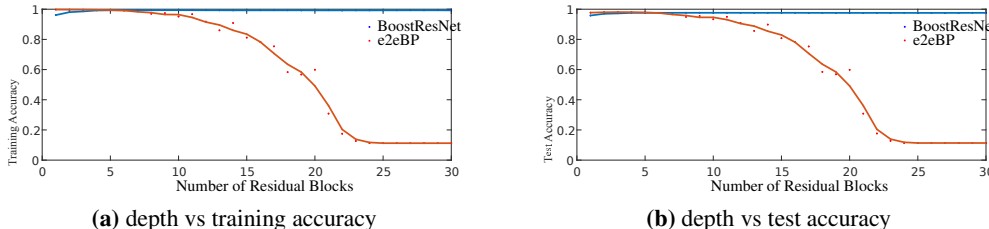

**(a)** depth vs training accuracy        **(b)** depth vs test accuracy

**Figure 2:** Comparison of *BoostResNet* (ours, blue) and *e2eBP* (baseline, red) on multilayer perceptron residual network on MNIST dataset.

**CNN-ResNet on SVHN** SVHN (Netzer et al., 2011) is a real-world image dataset, obtained from house numbers in Google Street View images. The dataset contains over 600,000 training images,

and about 20,000 test images. We fit a 50-layer, 25-residual-block CNN-ResNet using both *Boost-ResNet* and *e2eBP* (figure 3a). Each residual block is composed of a CNN using 15 $3 \times 3$ filters. We refine the result of *BoostResNet* by initializing the weights using the result of *BoostResNet* and run end-to-end back propagation (e2eBP). From figure 3a, our *BoostResNet* converges much faster (requires much fewer gradient updates) than *e2eBP*. The test accuracy of *BoostResNet* is comparable with *e2eBP*.

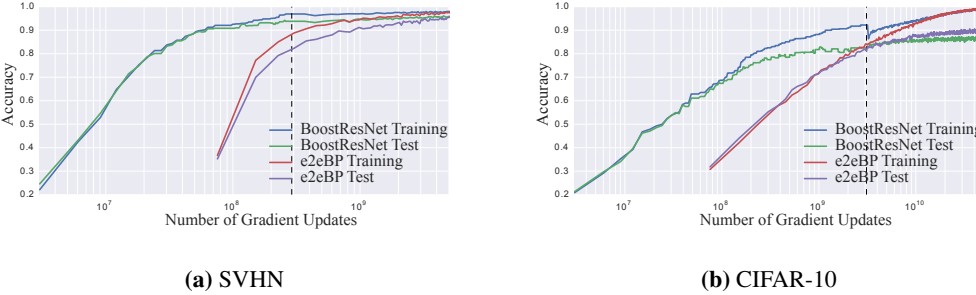

**(a)** SVHN                                     **(b)** CIFAR-10

**Figure 3:** Convergence performance comparison between *e2eBP* and *BoostResNet* on the SVHN and CIFAR-10 dataset. The vertical dotted line shows when BoostResNet training stopped, and we began refining the network with standard e2eBP training.

**CNN-ResNet on CIFAR-10** The CIFAR-10 dataset is a benchmark dataset composed of 10 classes of small images, such as animals and vehicles. It consists of 50,000 training images and 10,000 test images. We again fit a 50-layer, 25-residual-block CNN-ResNet using both *BoostResNet* and *e2eBP* (figure 3b). *BoostResNet* training converges to the optimal solution faster than *e2eBP*. Unlike in the previous two datasets, the efficiency of BoostResNet comes at a cost when training with CIFAR-10. We find that the test accuracy of the *e2eBP* refined *BoostResNet* to be slightly lower than that produced by *e2eBP*.

**Weak Learning Condition Check** The weak learning condition (Definition 4.1) inspired by learning theory is checked in Figure 4. The required better than random guessing edge $\gamma_t$ is depicted in Figure 4a, it is always greater than 0 and our weak learning condition is thus non-vacuous. In Figure 4b, the representations we learned using *BoostResNet* is increasingly better (for this classification task) as the depth increases.

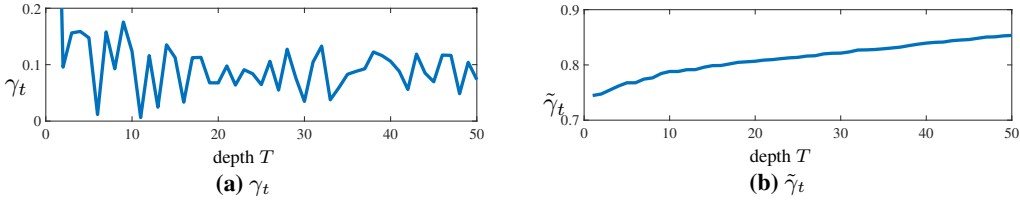

**(a)** $\gamma_t$                               **(b)** $\tilde{\gamma}_t$

**Figure 4:** Visualization of required larger than 0 edge $\gamma_t$ and edge for each residual block $\tilde{\gamma}_t$. The x-axis represents depth, and the y-axis represents $\gamma_t$ or $\tilde{\gamma}_t$ values. The plots are for a convolutional network composed of 50 residual blocks and trained on the SVHN dataset.

## 6 CONCLUSIONS AND FUTURE WORKS

Our proposed BoostResNet algorithm achieves exponentially decaying (with the depth $T$) training error under the weak learning condition. BoostResNet is much more computationally efficient compared to end-to-end back-propagation in deep ResNet. More importantly, the memory required by BoostResNet is trivial compared to end-to-end back-propagation. It is particularly beneficial given the limited GPU memory and large network depth. Our learning framework is natural for non-differentiable data. For instance, our learning framework is amenable to take weak learning oracles using tensor decomposition techniques. Tensor decomposition, a spectral learning framework with theoretical guarantees, is applied to learning one layer MLP in (Janzamin et al., 2015). We plan to extend our learning framework to non-differentiable data using general weak learning oracles.

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

# Appendix: Learning Deep ResNet Blocks Sequentially using Boosting Theory

## A  RELATED WORKS

### A.1  LOSS FUNCTION AND ARCHITECTURE SELECTION

In neural network optimization, there are many commonly-used loss functions and criteria, e.g., mean squared error, negative log likelihood, margin criterion, etc. There are extensive works (Girshick, 2015; Rubinstein & Kroese, 2013; Tygert et al., 2015) on selecting or modifying loss functions to prevent empirical difficulties such as exploding/vanishing gradients or slow learning (Balduzzi et al., 2017). However, there are no rigorous principles for selecting a loss function in general. Other works consider variations of the multilayer perceptron (MLP) or convolutional neural network (CNN) by adding identity skip connections (He et al., 2016), allowing information to bypass particular layers. However, no theoretical guarantees on the training error are provided despite breakthrough empirical successes. Hardt et al. (Hardt & Ma, 2016) have shown the advantage of identity loops in linear neural networks with theoretical justifications; however the linear setting is unrealistic in practice.

### A.2  LEARNING ALGORITHM DESIGN

There have been extensive works on improving BP (LeCun et al., 1989). For instance, momentum (Qian, 1999), Nesterov accelerated gradient (Nesterov, 1983), Adagrad (Duchi et al., 2011) and its extension Adadelta (Zeiler, 2012). Most recently, Adaptive Moment Estimation (Adam) (Kingma & Ba, 2014), a combination of momentum and Adagrad, has received substantial success in practice. All these methods are modifications of stochastic gradient descent (SGD), but our method only requires an arbitrary oracle, which does not necessarily need to be an SGD solver, that solves a relatively simple shallow neural network.

## B  PROOF FOR LEMMA 3.2: THE STRONG LEARNER IS A RESNET

*Proof.* In our algorithm, the input of the next module is the output of the current module

$$g_{t+1}(x) = f_t(g_t(x)) + g_t(x), \tag{8}$$

we thus obtain that each weak learning module is

$$h_t(x) = \alpha_{t+1}\mathbf{w}_{t+1}^\top(f_t(g_t(x)) + g_t(x)) - \alpha_t\mathbf{w}_t^\top g_t(x) \tag{9}$$

$$= \alpha_{t+1}\mathbf{w}_{t+1}^\top g_{t+1}(x) - \alpha_t\mathbf{w}_t^\top g_t(x), \tag{10}$$

and similarly

$$h_{t+1} = \alpha_{t+2}\mathbf{w}_{t+2}^\top g_{t+2}(x) - \alpha_{t+1}\mathbf{w}_{t+1}^\top g_{t+1}(x). \tag{11}$$

Therefore the sum over $h_t(x)$ and $h_{t+1}(x)$ is

$$h_t(x) + h_{t+1}(x) = \alpha_{t+2}\mathbf{w}_{t+2}^\top g_{t+2}(x) - \alpha_t\mathbf{w}_t^\top g_t(x) \tag{12}$$

And we further see that the weighted summation over all $h_t(x)$ is a telescoping sum

$$\sum_{t=1}^{T} h_t(x) = \alpha_{T+1}w_{T+1}^\top g_{T+1}(x) - \alpha_1 w_1^\top g_1(x) = \alpha_{T+1}w_{T+1}^\top g_{T+1}(x). \tag{13}$$

$\square$

## C  PROOF FOR THEOREM 4.2: BINARY CLASS TELESCOPING SUM BOOSTING THEORY

*Proof.* We will use a 0-1 loss to measure the training error. In our analysis, the 0-1 loss is bounded by exponential loss.

The training error is therefore bounded by

$$\Pr_{i \sim D_1} (p(\alpha_{T+1} w_{T+1}^\top g_{T+1}(x_i)) \neq y_i) \tag{14}$$

$$= \sum_{i=1}^{m} D_1(i) \mathbf{1}\{\tilde{\sigma}(\alpha_{T+1} w_{T+1}^\top g_{T+1}(x_i)) \neq y_i\} \tag{15}$$

$$= \sum_{i=1}^{m} D_1(i) \mathbf{1}\left\{ \tilde{\sigma}\left( \sum_{t=1}^{T} h_t(x_i) \right) \neq y_i \right\} \tag{16}$$

$$\leq \sum_{i=1}^{m} D_1(i) \exp\left\{ -y_i \sum_{t=1}^{T} h_t(x_i) \right\} \tag{17}$$

$$= \sum_{i=1}^{m} D_{T+1}(i) \prod_{t=1}^{T} Z_t \tag{18}$$

$$= \prod_{t=1}^{T} Z_t \tag{19}$$

where $Z_t = \sum\limits_{i=1}^{m} D_t(i) \exp\left(-y_i h_t(x_i)\right)$.

We choose $\alpha_{t+1}$ to minimize $Z_t$.

$$\frac{\partial Z_t}{\partial \alpha_{t+1}} = - \sum_{i=1}^{m} D_t(i) y_i o_{t+1} \exp\left(-y_i h_t(x_i)\right) \tag{20}$$

$$= -Z_t \sum_{i=1}^{m} D_{t+1}(i) y_i o_{t+1}(i) = 0 \tag{21}$$

Furthermore each learning module is bounded as we see in the following analysis. We obtain

$$Z_t = \sum_{i=1}^{m} D_t(i) e^{-y_i h_t(x_i)} \tag{22}$$

$$= \sum_{i=1}^{m} D_t(i) e^{-\alpha_{t+1} y_i o_{t+1}(x_i) + \alpha_t y_i o_t(x_i)} \tag{23}$$

$$\leq \sum_{i=1}^{m} D_t(i) e^{-\alpha_{t+1} y_i o_{t+1}(x_i)} \sum_{i=1}^{m} D_t(i) e^{\alpha_t y_i o_t(x_i)} \tag{24}$$

$$= \sum_{i=1}^{m} D_t(i) e^{-\alpha_{t+1} \frac{1 + y_i o_{t+1}(x_i)}{2} + \alpha_{t+1} \frac{1 - y_i o_{t+1}(x_i)}{2}} \sum_{i=1}^{m} D_t(i) e^{\alpha_t \frac{1 + y_i o_t(x_i)}{2} - \alpha_t \frac{1 - y_i o_t(x_i)}{2}} \tag{25}$$

$$\leq \sum_{i=1}^{m} D_t(i) \left( \frac{1 + y_i o_{t+1}(x_i)}{2} e^{-\alpha_{t+1}} + \frac{1 - y_i o_{t+1}(x_i)}{2} e^{\alpha_{t+1}} \right) . \tag{26}$$

$$\sum_{i=1}^{m} D_t(i) \left( \frac{1 + y_i o_t(x_i)}{2} e^{\alpha_t} + \frac{1 - y_i o_t(x_i)}{2} e^{-\alpha_t} \right) \tag{27}$$

$$= \sum_{i=1}^{m} D_t(i) \left( \frac{1 + y_i o_{t+1}(x_i)}{2} e^{-\alpha_{t+1}} + \frac{1 - y_i o_{t+1}(x_i)}{2} e^{\alpha_{t+1}} \right) \frac{e^{\alpha_t} + e^{-\alpha_t}}{2} \tag{28}$$

$$= \sum_{i=1}^{m} D_t(i) \left( \frac{e^{-\alpha_{t+1}} + e^{\alpha_{t+1}}}{2} + \frac{e^{-\alpha_{t+1}} - e^{\alpha_{t+1}}}{2} y_i o_{t+1}(x_i) \right) \frac{e^{\alpha_t} + e^{-\alpha_t}}{2} \tag{29}$$

$$= \left( \frac{e^{-\alpha_{t+1}} + e^{\alpha_{t+1}}}{2} + \frac{e^{-\alpha_{t+1}} - e^{\alpha_{t+1}}}{2} \tilde{\gamma}_t \right) \frac{e^{\alpha_t} + e^{-\alpha_t}}{2} \tag{30}$$

Equation (24) is due to the non-positive correlation between $\exp(-yo_{t+1}(x))$ and $\exp(yo_t(x))$. Jensen's inequality in Equation (27) holds only when $|y_i o_{t+1}(x_i)| \leq 1$ which is satisfied by the definition of the weak learning module.

The algorithm chooses $\alpha_{t+1}$ to minimize $Z_t$. We achieve an upper bound on $Z_t$, $\sqrt{\frac{1-\tilde{\gamma}_t^2}{1-\tilde{\gamma}_{t-1}^2}}$ by minimizing the bound in Equation (30)

$$Z_t \leq \left( \frac{e^{-\alpha_{t+1}} + e^{\alpha_{t+1}}}{2} + \frac{e^{-\alpha_{t+1}} - e^{\alpha_{t+1}}}{2} \tilde{\gamma}_t \right) \frac{e^{\alpha_t} + e^{-\alpha_t}}{2} \Bigg|_{\alpha_{t+1} = \frac{1}{2} \ln\left( \frac{1+\tilde{\gamma}_t}{1-\tilde{\gamma}_t} \right)} \tag{31}$$

$$= \sqrt{\frac{1-\tilde{\gamma}_t^2}{1-\tilde{\gamma}_{t-1}^2}} = \sqrt{1-\gamma_t^2} \tag{32}$$

Therefore over the $T$ modules, the training error is upper bounded as follows

$$\Pr_{i \sim D}\left(p(\alpha_{T+1} w_{T+1}^\top g_{T+1}(x_i))) \neq y_i\right) \leq \prod_{t=1}^T \sqrt{1-\gamma_t^2} \leq \prod_{t=1}^T \sqrt{1-\gamma^2} = \exp\left(-\frac{1}{2}T\gamma^2\right) \tag{33}$$

Overall, Algorithm 1 leads us to consistent learning of ResNet. $\qquad\square$

## D  PROOF FOR COROLLARY 4.3: GENERALIZATION BOUND

Rademacher complexity technique is powerful for measuring the complexity of $\mathcal{H}$ any family of functions $h : \mathcal{X} \to \mathbb{R}$, based on easiness of fitting any dataset using classifiers in $\mathcal{H}$ (where $\mathcal{X}$ is any space). Let $S = < x_1, \ldots, x_m >$ be a sample of $m$ points in $\mathcal{X}$. The empirical Rademacher complexity of $\mathcal{H}$ with respect to $S$ is defined to be

$$\mathcal{R}_S(\mathcal{H}) \stackrel{\text{def}}{=} \mathbb{E}_\sigma \left[ \sup_{h \in \mathcal{H}} \frac{1}{m} \sum_{i=1}^m \sigma_i h(x_i) \right] \tag{34}$$

where $\sigma$ is the Rademacher variable. The Rademacher complexity on $m$ data points drawn from distribution $\mathcal{D}$ is defined by

$$\mathcal{R}_m(\mathcal{H}) = \mathbb{E}_{S \sim \mathcal{D}}\left[ \mathcal{R}_S(\mathcal{H}) \right]. \tag{35}$$

**Proposition D.1.** *(Theorem 1 Cortes et al. (2014)) Let $\mathcal{H}$ be a hypothesis set admitting a decomposition $\mathcal{H} = \cup_{i=1}^l \mathcal{H}_i$ for some $l > 1$. $\mathcal{H}_i$ are distinct hypothesis sets. Let $S$ be a random sequence of $m$ points chosen independently from $\mathcal{X}$ according to some distribution $\mathcal{D}$. For $\theta > 0$ and any $H = \sum_{t=1}^T h_t$, with probability at least $1 - \delta$,*

$$\Pr_{\mathcal{D}}\left(yH(x) \leq 0\right) \leq \Pr_S\left(yH(x) \leq \theta\right) + \frac{4}{\theta} \sum_{t=1}^T \mathcal{R}_m(\mathcal{H}_{k_t}) + \frac{2}{\theta}\sqrt{\frac{\log l}{m}}$$

$$+ \sqrt{\left\lceil \frac{4}{\theta^2} \log\left(\frac{\theta^2 m}{\log l}\right)\right\rceil \frac{\log l}{m} + \frac{\log \frac{2}{\delta}}{2m}} \tag{36}$$

*for all $h_t \in \mathcal{H}_{k_t}$.*

**Lemma D.2.** *Let $\tilde{h} = \tilde{\mathbf{w}}^\top \tilde{\mathbf{f}}$, where $\tilde{\mathbf{w}} \in \mathbb{R}^n$, $\tilde{\mathbf{f}} \in \mathbb{R}^n$. Let $\tilde{\mathcal{H}}$ and $\tilde{\mathcal{F}}$ be two hypothesis sets, and $\tilde{h} \in \tilde{\mathcal{H}}$, $\tilde{\mathbf{f}}_j \in \tilde{\mathcal{F}}$, $\forall j \in [n]$. The Rademacher complexity of $\tilde{\mathcal{H}}$ and $\tilde{\mathcal{F}}$ with respect to $m$ points from $\mathcal{D}$ are related as follows*

$$\mathcal{R}_m(\tilde{\mathcal{H}}) = \|\tilde{\mathbf{w}}\|_1 \mathcal{R}_m(\tilde{\mathcal{F}}). \tag{37}$$

### D.1  RESNET MODULE HYPOTHESIS SPACE

Let $n$ be the number of channels in ResNet, i.e., the number of input or output neurons in a module $\mathbf{f}_t(\mathbf{g}_t(x))$. We have proved that ResNet is equivalent as

$$F(x) = \mathbf{w}^\top \sum_{t=1}^T \mathbf{f}(\mathbf{g}_t(x)) \tag{38}$$

We define the family of functions that each neuron $f_{t,j}, \forall j \in [n]$ belong to as

$$\mathcal{F}_t = \{x \to \mathbf{u}_{t-1,j}(\sigma \circ \mathbf{f}_{t-1})(x) : \mathbf{u}_{t-1,j} \in \mathbb{R}^n, \|\mathbf{u}_{t-1,j}\|_1 \leq \Lambda_{t,t-1}, \mathbf{f}_{t-1,i} \in \mathcal{F}_{t-1}\} \quad (39)$$

where $\mathbf{u}_{t-1,j}$ denotes the vector of weights for connections from unit $j$ to a lower layer $t-1$, $\sigma \circ \mathbf{f}_{t-1}$ denotes element-wise nonlinear transformation on $\mathbf{f}_{t-1}$. The output layer of each module is connected to the output layer of previous module. We consider 1-layer modules for convenience of analysis.

Therefore in ResNet with probability at least $1 - \delta$,

$$\Pr_{\mathcal{D}}(yF(x) \leq 0) \leq \Pr_{S}(yF(x) \leq \theta) + \frac{4}{\theta}\sum_{t=1}^{T}\|\mathbf{w}\|_1 \mathcal{R}_m(\mathcal{F}_t) + \frac{2}{\theta}\sqrt{\frac{\log T}{m}}$$

$$+ \sqrt{\lceil \frac{4}{\theta^2}\log\left(\frac{\theta^2 m}{\log T}\right)\rceil\frac{\log T}{m} + \frac{\log\frac{2}{\delta}}{2m}} \quad (40)$$

for all $f_t \in \mathcal{F}_t$.

Define the maximum infinity norm over samples as $r_\infty \overset{\text{def}}{=} \mathbb{E}_{S\sim\mathcal{D}}\left[\max_{i\in[m]}\|x_i\|_\infty\right]$ and the product of $l_1$ norm bound on weights as $\Lambda_t \overset{\text{def}}{=} \prod_{t'=1}^{t}2\Lambda_{t',t'-1}$. According to lemma 2 of Cortes et al. (2016), the empirical Rademacher complexity is bounded as a function of $r_\infty$, $\Lambda_t$ and $n$:

$$\mathcal{R}_m(\mathcal{F}_t) \leq r_\infty\Lambda_t\sqrt{\frac{\log(2n)}{2m}} \quad (41)$$

Overall, with probability at least $1 - \delta$,

$$\Pr_{\mathcal{D}}(yF(x) \leq 0) \leq \Pr_{S}(yF(x) \leq \theta) + \frac{4\|\mathbf{w}\|_1 r_\infty\sqrt{\frac{\log(2n)}{2m}}}{\theta}\sum_{t=1}^{T}\Lambda_t$$

$$+ \frac{2}{\theta}\sqrt{\frac{\log T}{m}} + \sqrt{\lceil\frac{4}{\theta^2}\log\left(\frac{\theta^2 m}{\log T}\right)\rceil\frac{\log T}{m} + \frac{\log\frac{2}{\delta}}{2m}} \quad (42)$$

for all $f_t \in \mathcal{F}_t$.

## E    PROOF FOR THEOREM E: MARGIN AND GENERALIZATION BOUND

**Theorem E.1.** [ ***Generalization error bound*** ] *Given algorithm 1, the fraction of training examples with margin at most $\theta$ is at most $(1 + \frac{2}{\frac{1}{\sqrt{\gamma}_{T+1}}-1})^{\frac{\theta}{2}}\exp(-\frac{1}{2}\gamma^2 T)$. And the generalization error* $\Pr_D(yF(x) \leq 0)$ *satisfies*

$$\Pr_{D}(yF(x) \leq 0) \leq (1 + \frac{2}{\frac{1}{\overline{\gamma}_{T+1}}-1})^{\frac{\theta}{2}}\exp(-\frac{1}{2}\gamma^2 T)$$

$$+ \frac{4C_0 r_\infty}{\theta}\sqrt{\frac{\log(2n)}{2m}}\sum_{t=1}^{T}\Lambda_t + \frac{2}{\theta}\sqrt{\frac{\log T}{m}} + \beta(\theta, m, T, \delta) \quad (43)$$

*with probability at least $1 - \delta$ for $\beta(\theta, m, T, \delta) \overset{\text{def}}{=} \sqrt{\lceil\frac{4}{\theta^2}\log\left(\frac{\theta^2 m}{\log T}\right)\rceil\frac{\log T}{m} + \frac{\log\frac{2}{\delta}}{2m}}$.*

Now the proof for Theorem E is the following.

*Proof.* The fraction of examples in sample set $S$ being smaller than $\theta$ is bounded

$$\Pr_S(yF(x) \leq \theta) \leq \frac{1}{m} \sum_{i=1}^{m} \mathbf{1}\{y_i F(x_i) \leq \theta\} \tag{44}$$

$$= \frac{1}{m} \sum_{i=1}^{m} \mathbf{1}\{y_i \sum_{t=1}^{T} h_t(x_i) \leq \theta\alpha_{T+1}\} \tag{45}$$

$$\leq \frac{1}{m} \sum_{i=1}^{m} \exp(-y_i \sum_{t=1}^{T} h_t(x_i) + \theta\alpha_{T+1}) \tag{46}$$

$$= \exp(\theta\alpha_{T+1}) \frac{1}{m} \sum_{i=1}^{m} \exp(-y_i \sum_{t=1}^{T} h_t(x_i)) \tag{47}$$

$$= \exp(\theta\alpha_{T+1}) \prod_{t=1}^{T} Z_t \tag{48}$$

To bound $\exp(\theta\alpha_{T+1}) = \sqrt{(\frac{1+\tilde{\gamma}_{T+1}}{1-\tilde{\gamma}_{T+1}})^\theta}$, we first bound $\tilde{\gamma}_{T+1}$: We know that $\sum_{t=1}^{T} \prod_{t'=t+1}^{T}(1 - \gamma_{t'}^2)\gamma_t^2 \leq (1-\gamma^2)^{T-t}\gamma^2$ for all $\forall \gamma_t \geq \gamma^2 + \epsilon$ if $\gamma^2 \geq \frac{1-\epsilon}{2}$. Therefore $\forall \gamma_t \geq \gamma^2 + \epsilon$ and $\gamma^2 \geq \frac{1-\epsilon}{2}$

$$\tilde{\gamma}_{T+1}^2 = (1 - \gamma_T^2)\tilde{\gamma}_T^2 + \gamma_T^2 \tag{49}$$

$$= \sum_{t=1}^{T} \prod_{t'=t+1}^{T}(1 - \gamma_{t'}^2)\gamma_t^2 + \prod_{t=1}^{T}(1 - \gamma_t^2)\tilde{\gamma}_1^2 \tag{50}$$

$$\leq \sum_{t=1}^{T}(1 - \gamma^2)^{T-t}\gamma^2 + (1 - \gamma^2)^T\tilde{\gamma}_1^2 \tag{51}$$

$$= \sum_{t=0}^{T-1}(1 - \gamma^2)^t\gamma^2 + (1 - \gamma^2)^T\tilde{\gamma}_1^2 \tag{52}$$

$$= 1 - (1 - \gamma^2)^T + (1 - \gamma^2)^T\tilde{\gamma}_1^2 \tag{53}$$

$$= 1 - (1 - \tilde{\gamma}_1^2)(1 - \gamma^2)^T \tag{54}$$

Therefore

$$\Pr_S(yF(x) \leq \theta) \leq \exp(\theta\alpha_{T+1}) \prod_{t=1}^{T} Z_t \tag{55}$$

$$= (\frac{1 + \tilde{\gamma}_{T+1}}{1 - \tilde{\gamma}_{T+1}})^{\frac{\theta}{2}} \prod_{t=1}^{T} Z_t \tag{56}$$

$$= (\frac{1 + \tilde{\gamma}_{T+1}}{1 - \tilde{\gamma}_{T+1}})^{\frac{\theta}{2}} \prod_{t=1}^{T} \sqrt{1 - \gamma_t^2} \tag{57}$$

$$= (1 + \frac{2}{\frac{1}{\tilde{\gamma}_{T+1}} - 1})^{\frac{\theta}{2}} \exp(-\frac{1}{2}\gamma^2 T) \tag{58}$$

$$\leq (1 + \frac{2}{\frac{1}{\sqrt{1-(1-\tilde{\gamma}_1^2)(1-\gamma^2)^T}} - 1})^{\frac{\theta}{2}} \exp(-\frac{1}{2}\gamma^2 T) \tag{59}$$

As $T \to \infty$, $\Pr_S(yF(x) \leq \theta) \leq 0$ as $\exp(-\frac{1}{2}\gamma^2 T)$ decays faster than $(1 + \frac{2}{\frac{1}{\sqrt{1-(1-\tilde{\gamma}_1^2)(1-\gamma^2)^T}} - 1})^{\frac{\theta}{2}}$.

$\square$

# F    TELESCOPING SUM BOOSTING FOR MULTI-CALSS CLASSIFICATION

Recall that the weak module classifier is defined as

$$h_t(x) = \alpha_{t+1}o_{t+1}(x) - \alpha_t o_t(x) \in \mathbb{R}^C, \tag{60}$$

where $o_t(x) \in \Delta^{C-1}$.

The weak learning condition for multi-class classification is different from the binary classification stated in the previous section, although minimal demands placed on the weak module classifier require prediction better than random on any distribution over the training set intuitively.

We now define the weak learning condition. It is again inspired by the slightly better than random idea, but requires a more sophisticated analysis in the multi-class setting.

## F.1    COST MATRIX

In order to characterize the training error, we introduce the cost matrix $\mathbf{C} \in \mathbb{R}^{m \times C}$ where each row denote the cost incurred by classifying that example into one of the $C$ categories. We will bound the training error using exponential loss, and under the exponential loss function defined as in Definition G.1, the optimal cost function used for best possible training error is therefore determined.

**Lemma F.1.** *The optimal cost function under the exponential loss is*

$$\mathbf{C}_t(i,l) = \begin{cases} \exp\left(s_t(x_i, l) - s_t(x_i, y_i)\right) & \text{if } l \neq y_i \\ -\sum_{l' \neq y_i} \exp\left(s_t(x_i, l') - s_t(x_i, y_i)\right) & \text{if } l = y_i \end{cases} \tag{61}$$

*where $s_t(x) = \sum_{\tau=1}^{t} h_\tau(x)$.*

## F.2    WEAK LEARNING CONDITION

**Definition F.2.** *Let $\tilde{\gamma}_{t+1} = \dfrac{-\sum_{i=1}^{m} <\mathbf{C}_t(i,:), o_{t+1}(x_i)>}{\sum_{i=1}^{m} \sum_{l \neq y_i} \mathbf{C}_t(i,l)}$ and $\tilde{\gamma}_t = \dfrac{-\sum_{i=1}^{m} <\mathbf{C}_{t-1}(i,:), o_t(x_i)>}{\sum_{i=1}^{m} \sum_{l \neq y_i} \mathbf{C}_{t-1}(i,l)}$. A multi-class weak module classifier $h_t(x) = \alpha_{t+1}o_{t+1}(x) - \alpha_t o_t(x)$ satisfies the $\gamma$-weak learning condition if $\dfrac{\tilde{\gamma}_{t+1}^2 - \tilde{\gamma}_t^2}{1 - \tilde{\gamma}_t^2} \geq \gamma^2 > 0$, and $Cov(<\mathbf{C}_t(i,:), o_{t+1}(x_i)>, <\mathbf{C}_t(i,:), o_{t+1}(x_i)>) \geq 0$.*

We propose a novel learning algorithm using the optimal edge-over-random cost function for training ResNet under multi-class classification task as in Algorithm 3.

**Theorem F.3.** *The training error of a $T$-module ResNet using Algorithm 3and 4 decays exponentially with the depth of the ResNet $T$,*

$$\frac{C-1}{m} \sum_{i=1}^{m} L_\eta^{exp}(s_T(x_i)) \leq (C-1)e^{-\frac{1}{2}T\gamma^2} \tag{62}$$

*if the weak module classifier $h_t(x)$ satisfies the $\gamma$-weak learning condition $\forall t \in [T]$.*

The exponential loss function defined as in Definition G.1

---

**Algorithm 3** BoostResNet: telescoping sum boosting for multi-class classification

---

**Input:** Given $(x_1, y_1), \ldots (x_m, y_m)$ where $y_i \in \mathcal{Y} = \{1, \ldots, C\}$ and a threshold $\gamma$
**Output:** $\{f_t(\cdot), \forall t\}$ and $W_{T+1}$         $\triangleright$ Discard $\mathbf{w}_{t+1}, \forall t \neq T$
 1: Initialize $t \leftarrow 0$, $\tilde{\gamma}_0 \leftarrow 1$, $\alpha_0 \leftarrow 0$, $o_0 \leftarrow \mathbf{0} \in \mathbb{R}^C$, $s_0(x_i, l) = 0, \forall i \in [m], l \in \mathcal{Y}$
 2: Initialize cost function $\mathbf{C}_0(i, l) \leftarrow \begin{cases} 1 & \text{if } l \neq y_i \\ 1 - C & \text{if } l = y_i \end{cases}$
 3: **while** $\gamma_t > \gamma$ **do**
 4:     $f_t(\cdot), \alpha_{t+1}, W_{t+1}, o_{t+1}(x) \leftarrow$ Algorithm 4$(g_t(x), \mathbf{C}_t, o_t(x), \alpha_t)$
 5:     Compute $\gamma_t \leftarrow \sqrt{\frac{\tilde{\gamma}_{t+1}^2 - \tilde{\gamma}_t^2}{1 - \tilde{\gamma}_t^2}}$         $\triangleright$ where $\tilde{\gamma}_{t+1} \leftarrow \dfrac{-\sum\limits_{i=1}^{m} \mathbf{C}_t(i,:) \cdot o_{t+1}(x_i)}{\sum\limits_{i=1}^{m} \sum\limits_{l \neq y_i} \mathbf{C}_t(i,l)}$
 6:     Update $s_{t+1}(x_i, l) \leftarrow s_t(x_i, l) + h_t(x_i, l) \triangleright$ where $h_t(x_i, l) = \alpha_{t+1} o_{t+1}(x_i, l) - \alpha_t o_t(x_i, l)$
 7:     Update cost function $\mathbf{C}_{t+1}(i, l) \leftarrow \begin{cases} e^{s_{t+1}(x_i,l) - s_{t+1}(x_i,y_i)} & \text{if } l \neq y_i \\ -\sum\limits_{l' \neq y_i} e^{s_{t+1}(x_i,l') - s_{t+1}(x_i,y_i)} & \text{if } l = y_i \end{cases}$
 8:     $t \leftarrow t + 1$
 9: **end while**
10: $T \leftarrow t - 1$

---

**Algorithm 4** BoostResNet: oracle implementation for training a ResNet module (multi-class)

---

**Input:** $g_t(x), s_t, o_t(x)$ and $\alpha_t$
**Output:** $f_t(\cdot), \alpha_{t+1}, W_{t+1}$ and $o_{t+1}(x)$
 1: $(f_t, \alpha_{t+1}, W_{t+1}) \leftarrow \arg \min\limits_{(f,\alpha,V)} \sum\limits_{i=1}^{m} \sum\limits_{l \neq y_i} e^{\alpha V^\top [f(g_t(x_i),l) - f(g_t(x_i),y_i) + g_t(x_i,l) - g_t(x_i,y_i)]}$
 2: $o_{t+1}(x) \leftarrow W_{t+1}^\top [f_t(g_t(x)) + g_t(x)]$

---

### F.3 ORACLE IMPLEMENTATION

We implement an oracle to minimize $Z_t \overset{\text{def}}{=} \sum\limits_{i=1}^{m} \sum\limits_{l \neq y_i} e^{s_t(x_i,l) - s_t(x_i,y_i)} e^{h_t(x_i,l) - h_t(x_i,y_i)}$ given current state $s_t$ and hypothesis module $o_t(x)$. Therefore minimizing $Z_t$ is equivalent to the following.

$$\min_{(f,\alpha,V)} \sum_{i=1}^{m} \sum_{l \neq y_i} e^{s_t(x_i,l) - s_t(x_i,y_i)} e^{-\alpha_t(o_t(x_i,l) - o_t(x_i,y_i))} e^{\alpha V^\top [f(g_t(x_i),l) - f(g_t(x_i),y_i) + g_t(x_i,l) - g_t(x_i,y_i)]} \tag{63}$$

$$\equiv \min_{(f,\alpha,V)} \sum_{i=1}^{m} \sum_{l \neq y_i} e^{\alpha V^\top [f(g_t(x_i),l) - f(g_t(x_i),y_i) + g_t(x_i,l) - g_t(x_i,y_i)]} \tag{64}$$

$$\equiv \min_{\alpha,f,v} \sum_{i=1}^{m} e^{-\alpha v^\top [f(x_i,y_i) + g_t(x_i,y_i)]} \sum_{l \neq y_i} e^{\alpha v^\top [f(x_i,l) + g_t(x_i,l)]} \tag{65}$$

## G PROOF FOR THEOREM F.3 MULTICLASS BOOSTING THEORY

*Proof.* To characterize the training error, we use the exponential loss function

**Definition G.1.** *Define loss function for a multiclass hypothesis $H(x_i)$ on a sample $(x_i, y_i)$ as*

$$L_\eta^{exp}(H(x_i), y_i) = \sum_{l \neq y_i} \exp\left((H(x_i,l) - H(x_i,y_i))\right). \tag{66}$$

Define the accumulated weak learner $s_t(x_i, l) = \sum\limits_{t'=1}^{t} h_{t'}(x_i, l)$ and the loss $Z_t = \sum\limits_{i=1}^{m} \sum\limits_{l \neq y_i} \exp(s_t(x_i, l) - s_t(x_i, y_i)) \exp(h_t(x_i, l) - h_t(x_i, y_i))$.

Recall that $s_t(x_i, l) = \sum_{t'=1}^{t} h_{t'}(x_i, l) = \alpha_{t+1} W_{t+1}^\top g_{t+1}(x_i)$, the loss for a $T$-module multiclass ResNet is thus

$$\Pr_{i \sim D_1} (p(\alpha_{T+1} W_{T+1}^\top g_{T+1}(x_i)) \neq y_i) \leq \frac{1}{m} \sum_{i=1}^{m} L_\eta^{exp}(s_T(x_i)) \tag{67}$$

$$\leq \frac{1}{m} \sum_{i=1}^{m} \sum_{l \neq y_i} \exp\left(\eta(s_T(x_i, l) - s_T(x_i, y_i))\right) \tag{68}$$

$$\leq \frac{1}{m} Z_T \tag{69}$$

$$= \prod_{t=1}^{T} \frac{Z_t}{Z_{t-1}} \tag{70}$$

Note that $Z_0 = \frac{1}{m}$ as the initial accumulated weak learners $s_0(x_i, l) = 0$.

The loss fraction between module $t$ and $t-1$, $\frac{Z_t}{Z_{t-1}}$, is related to $Z_t - Z_{t-1}$ as $\frac{Z_t}{Z_{t-1}} = \frac{Z_t - Z_{t-1}}{Z_{t-1}} + 1$. The $Z_t$ is bounded

$$Z_t = \sum_{i=1}^{m} \sum_{l \neq y_i} \exp(s_t(x_i, l) - s_t(i, y_i) + h_t(x_i, l) - h_t(x_i, y_i)) \tag{71}$$

$$\leq \sum_{i=1}^{m} \sum_{l \neq y_i} e^{s_t(x_i,l) - s_t(x_i,y_i)} e^{\alpha_{t+1} o_{t+1}(x_i,l) - \alpha_{t+1} o_{t+1}(x_i,y_i)} \sum_{i=1}^{m} \sum_{l \neq y_i} e^{s_t(x_i,l) - s_t(x_i,y_i)} e^{-\alpha_t o_t(x_i,l) + \alpha_t o_t(x_i,y_i)} \tag{72}$$

$$\leq \sum_{i=1}^{m} \sum_{l \neq y_i} e^{s_t(x_i,l) - s_t(x_i,y_i)} \left( \frac{e^{-\alpha_{t+1}} + e^{\alpha_{t+1}}}{2} + \frac{e^{-\alpha_{t+1}} - e^{\alpha_{t+1}}}{2} (o_{t+1}(x_i, y_i) - o_{t+1}(x_i, l)) \right)$$
$$\sum_{i=1}^{m} \sum_{l \neq y_i} e^{s_{t-1}(x_i,l) - s_{t-1}(x_i,y_i)} \left( \frac{e^{\alpha_t} + e^{-\alpha_t}}{2} \right) \tag{73}$$

$$= \left( \frac{e^{-\alpha_{t+1}} + e^{\alpha_{t+1}} - 2}{2} Z_{t-1} + \frac{e^{\alpha_{t+1}} - e^{-\alpha_{t+1}}}{2} \sum_{i=1}^{m} < C_t(x_i, :), o_{t+1}(x_i, :) > \right) \left( \frac{e^{\alpha_t} + e^{-\alpha_t}}{2} \right) \tag{74}$$

$$\leq \left( \frac{e^{-\alpha_{t+1}} + e^{\alpha_{t+1}} - 2}{2} Z_{t-1} + \frac{e^{\alpha_{t+1}} - e^{-\alpha_{t+1}}}{2} \sum_{i=1}^{m} < C_t(x_i, :), U_{\tilde{\gamma}_t}(x_i, :) > \right) \left( \frac{e^{\alpha_t} + e^{-\alpha_t}}{2} \right) \tag{75}$$

$$= \left( \frac{e^{-\alpha_{t+1}} + e^{\alpha_{t+1}} - 2}{2} Z_{t-1} + \frac{e^{\alpha_{t+1}} - e^{-\alpha_{t+1}}}{2} (-\tilde{\gamma}_t) Z_{t-1} \right) \left( \frac{e^{\alpha_t} + e^{-\alpha_t}}{2} \right) \tag{76}$$

Therefore

$$\frac{Z_t}{Z_{t-1}} \leq \left( \frac{e^{-\alpha_{t+1}} + e^{\alpha_{t+1}}}{2} + \frac{e^{-\alpha_{t+1}} - e^{\alpha_{t+1}}}{2} \tilde{\gamma}_t \right) \left( \frac{e^{\alpha_t} + e^{-\alpha_t}}{2} \right) \tag{77}$$

The algorithm chooses $\alpha_{t+1}$ to minimize $Z_t$. We achieve an upper bound on $Z_t$, $\sqrt{\frac{1 - \tilde{\gamma}_t^2}{1 - \tilde{\gamma}_{t-1}^2}}$

by minimizing the bound in Equation (77)

$$Z_t \le \left( \frac{e^{-\alpha_{t+1}} + e^{\alpha_{t+1}}}{2} + \frac{e^{-\alpha_{t+1}} - e^{\alpha_{t+1}}}{2} \tilde{\gamma}_t \right) \frac{e^{\alpha_t} + e^{-\alpha_t}}{2} \bigg|_{\alpha_{t+1} = \frac{1}{2} \ln \left( \frac{1 + \tilde{\gamma}_t}{1 - \tilde{\gamma}_t} \right)} \quad (78)$$

$$= \sqrt{\frac{1 - \tilde{\gamma}_t^2}{1 - \tilde{\gamma}_{t-1}^2}} = \sqrt{1 - \gamma_t^2} \quad (79)$$

Therefore over the $T$ modules, the training error is upper bounded as follows

$$\Pr_{i \sim D} \left( p(\alpha_{T+1} w_{T+1}^\top g_{T+1}(x_i)) \ne y_i \right) \le \prod_{t=1}^T \sqrt{1 - \gamma_t^2} \le \prod_{t=1}^T \sqrt{1 - \gamma^2} = \exp \left( -\frac{1}{2} T \gamma^2 \right) \quad (80)$$

Overall, Algorithm 3 and 4 leads us to consistent learning of ResNet. $\qquad\square$

## H EXPERIMENTS

### H.1 TRAINING ERROR DEGRADATION OF E2EBP ON RESNET

We investigate e2eBP training performance on various depth ResNet. Surprisingly, we observe a training error degradation for *e2eBP* although the ResNet's identity loop is supposed to alleviate this problem. Despite the presence of identity loops, the *e2eBP* eventually is susceptible to spurious local optima. This phenomenon is explored further in Figures 5a and 5b, which respectively show how training and test accuracies vary throughout the fitting process. Our proposed sequential training procedure, *BoostResNet*, relieves gradient instability issues, and continues to perform well as depth increases.

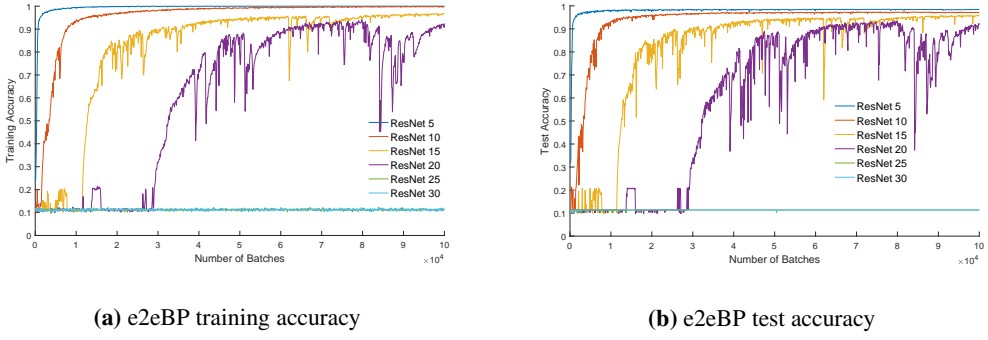

**(a)** e2eBP training accuracy          **(b)** e2eBP test accuracy

**Figure 5:** Convergence of *e2eBP* (baseline) on multilayer perceptron residual network (of various depths) on MNIST dataset.

### H.2 SVHN AND CIFAR-10 PERFORMANCE RESULTS

Besides e2eBP, we also experiment with standard boosting (AdaBoost.MM Mukherjee & Schapire (2013)), as another baseline, of convolutional modules. In this experiment, each weak learner is a residual block of the ResNet, paired with a classification layer. We do 25 rounds of AdaBoost.MM and train each weak learner to convergence.

Table 1 and table 2 exhibit a comparison of BoostResNet, e2eBP and AdaBoost performance on SVHN and CIFAR-10 dataset respectively.

On SVHN dataset, the advantage of BoostResNet over e2eBP is obvious. Using $3 \times 10^8$ number of gradient updates, BoostResNet achieves $93.8\%$ test accuracy whereas e2eBP obtains a test accuracy of $83\%$. The training and test accuracies of SVHN are listed in Table 1. BoostResNet training allows the model to train much faster than end-to-end training, and still achieves the same test accuracy when refined with *e2eBP*. To list the hyperparameters we use in our BoostResNet training

after searching over candidate hyperparamters, we choose learning rate to be 0.004 with a $9 \times 10^{-5}$ learning rate decay. The gamma threshold is set to be 0.001 and the initial gamma value on SVHN is 0.75.

On CIFAR-10 dataset, the main advantage of BoostResNet over e2eBP is the speed of training. BoostResNet refined with e2eBP obtains comparable results with e2eBP. This is because we are using a suboptimal architecture of ResNet which overfits the CIFAR-10 dataset. AdaBoost, on the other hand, is known to be resistant to overfitting. Therefore, AdaBoost achieves the highest test accuracy on CIFAR-10. To list the hyperparameters we use in our BoostResNet training after searching over candidate hyperparamters, we choose learning rate to be 0.014 with a $3.46 \times 10^{-5}$ learning rate decay. The gamma threshold is set to be 0.007 and the initial gamma value on CIFAR-10 is 0.93.

| TRAINING: | BOOSTRESNET | E2EBP | BOOSTRESNET+E2EBP | E2EBP | ADABOOST |
|---|---|---|---|---|---|
| NGU | $3 \times 10^8$ | $3 \times 10^8$ | $2 \times 10^{10}$ | $2 \times 10^{10}$ | $1.5 \times 10^9$ |
| TRAIN | 96.9% | 85% | 98.8% | 98.8% | 95.6% |
| TEST | 93.8% | 83% | 96.8% | 96.8% | 92.3% |

Table 1: **Accuracies of SVHN task.** NGU is the number of gradient updates taken by the algorithm in training.

| TRAINING: | BOOSTRESNET | E2EBP | BOOSTRESNET+E2EBP | E2EBP | ADABOOST |
|---|---|---|---|---|---|
| NGU | $3 \times 10^9$ | $3 \times 10^9$ | $1 \times 10^{11}$ | $1 \times 10^{11}$ | $1.5 \times 10^{10}$ |
| TRAIN | 92.1% | 82% | 99.6% | 99.7% | 95.6% |
| TEST | 82.1% | 80% | 88.1% | 90.0% | 92.3% |

Table 2: **Accuracies of CIFAR-10 task.** NGU is the number of gradient updates taken by the algorithm in training.

