# OpenReview forum: "Learning Deep ResNet Blocks Sequentially using Boosting Theory"
_ICLR.cc/2018/Conference — Reject_

### Official Review · AnonReviewer3 · 2017-11-27
**An ineresting approach to boosting residual networks but problems from NIPS reviews are still not resolved.**

**Rating:** 5
**Confidence:** 4

**Review:**

Disclaimer: I reviewed this paper for NIPS as well and many of comments made by reviewers at that time still apply to this version of the paper as well, although presentation has overall improved.

The paper presents a boosting-style algorithm for training deep residual networks. Convergence analysis for training error is presented and analysis of generalization ability is also provided. Paper concludes with some experimental results.

The main contribution of this work is interpretation of ResNet as a telescoping sum of differences between the intermediate layers and treating these differences as weak learners that are then boosted. This indeed appears to an interesting insight about ResNet training.

On the other hand, one of the main objections during NIPS reviews was the relation of this work to work of Cortes et al. on Adanet. In particular, generalization bounds presented in this work are results taken from that paper (which authors admit). What is less clear is the distinction between the algorithmic approaches which makes it hard to judge the novelty of this work. There is a paragraph at the end of section 2 but it seems rather vague.

One other objection during NIPS reviews was experimental setup explanation of which is omitted from the current version. In particular, same learning rate and mini-batch size was used both for boosting and backprop algorithms which seems strange since boosting is supposed to train much smaller classifiers.

Another concern is practicality of the proposed method which seems to require maintaining explicit distribution over all examples which would not be practical for modern datasets where NNs are typically applied.

---

> ### Author Response · Authors · 2017-12-11
> **The concerns from NIPS reviews were addressed in the ICLR submission**
>
> Thank you very much for your review for NIPS and ICLR. We did major modifications to the NIPS version of the paper to clarify the misunderstandings from the NIPS reviewers, which we didn't have a chance to address during NIPS rebuttal as we couldn't see all reviews due to the technical problems. We had rerun all experiments and had done hyperparameter optimization for each algorithm in the submitted ICLR version.
>
> 1. To respond to your concern “What is less clear is the distinction between the algorithmic approaches which makes it hard to judge the novelty of this work. There is a paragraph at the end of section 2 but it seems rather vague.”
>
> - Section 2 does not talk about the algorithm. It only serves as an introduction of preliminaries to prepare the readers for resnet and boosting. Our algorithm is different from Cortes et. al as discussed in details in section 1.2. Please take a look at section 1.2 (at the bottom of page 2).
>
> 2. To respond to your concern “One other objection during NIPS reviews was experimental setup explanation of which is omitted from the current version. In particular, same learning rate and mini-batch size was used both for boosting and backprop algorithms which seems strange since boosting is supposed to train much smaller classifiers.”
>
> - Our parameters are specified in the appendix and were optimized for performance.  We experimented with various learning parameters for both the e2e resnet and the boostresnet. In boostresnet, we found that the most important hyperparameters were those that govern when the algorithm stops training the current module and begins training its successor. We also found that a standard resnet, to its credit, is quite robust to hyperparameters, namely learning rate and learning rate decay, provided that we used an optimization procedure that automatically modulated these values (as mentioned, we used Adam). Changing these hyperparameters had a negligible affect on e2e model accuracy.
>
> 3. To respond to your concern “Another concern is practicality of the proposed method which seems to require maintaining explicit distribution over all examples which would not be practical for modern datasets where NNs are typically applied.”
>
> - We don’t understand this comment---the additional memory requirement is just number of classes * number of examples which would be about 100MB on ImageNet 20K.  Modern machines typically have a factor of 100 more RAM.
>
> We hope the reviewer could kindly reconsider the score and decision after reading our clarifications.

---

> > ### Comment · AnonReviewer3 · 2017-12-19
> > **reply**
> >
> > Thank you for you clarifications.
> >
> > 1. I was referring to section 1.2 (not 2). I find there just the following sentences: "Therefore, to obtain low training error guarantee, AdaNet maps the feature vectors (hidden layer representations) to a classifier space and boosts the weak classifiers. Our BoostResNet, instead, boosts representations (feature vectors) over multiple channels, and therefore produces a less “bushy” architecture."
> > Given connections between two papers, I think comparison with AdaNet deserves at least its own subsection where some of the notions in this paragraph are much clearer define.
> > For instance, what is the difference between mapping features to classifier space and boosting representations? Isn't mapping something to classifier space just another feature representation? And if there is any difference, why one is better than other?
> > What does "bushy" mean? Why or when "less" bushy is better?
> > Overall, is Adanet a special case of BoostResNet? Is it the other way around? are they uncomparable? Right now it is clear to me that theory for the two is the same and both of them boost NNs but the difference between the two need to be highlighted more.
> >
> > 2. Appendix H does not say that you experimented with various hparams. It just states the ones you used. If you experimented with different ones then please explain how you selected these particular ones.
> >
> > 3. I meant that to updated distribution one needs to compute partition function. This seems to be an expensive step since it basically requires a pass over data.

---

> > > ### Author Response · Authors · 2017-12-19
> > > **Clarify reviewer's concern -- compare with Adanet, and other minor clarifications**
> > >
> > > Thank you again for your feedback and questions. We would like to clarify some issues as follows.
> > >
> > > 1. The theory between the two is different. We emphasize that traditional boosting doesn't work in the Resnet setting. Traditional boosting ensembles "estimated score functions" (or even estimated labels) from weak learners. Our boosting ensembles "features" (representation from lower level layers). There is no boosting theory for ensembling features.
> > >
> > > We are able to boost features by developing this new "telescoping-sum boosting" framework, one of our main contributions.  We come up with the new weak learning condition for the telescoping-sum boosting framework. The algorithm is also very different from Adanet. These are explained in details in section 3 and 4 (the 8 page limit makes it hard to add another section to compare with Adanet, given that Adanet is so different from our algorithm). We rewrote the two sections (section 3 and 4) to make it clearer after receiving your comments from the nips submission.
> > >
> > > By "Bushy" we mean the following: In Adanet, features (representations) from each lower layer have to be fed into a classifier (in other words,  be transferred to score function in the label space).  This is because Adanet uses traditional boosting, which ensembles score functions or labels.  Therefore, the top classifier in Adanet has to be connected to all lower layers, making the structure bushy.  Therefore Adanet chooses their own structure during learning, and their boosting theory does not necessarily work for a Resnet structure.
> > >
> > > Our contribution does not limit to explaining Resnet in the Boosting framework, we have also developed a new boosting framework for many other relevant tasks.
> > >
> > > 2. Hyperparameters were found via random search, selected for highest accuracy on a validation set. We will add this to our paper. Thank you for your suggestion.
> > >
> > > 3. This is done through a cost function, explained in the paper (equation (61)),  and it is inexpensive to update according to our experiments.

---

> > > > ### Comment · AnonReviewer3 · 2018-01-05
> > > > **reply**
> > > >
> > > > 1. First of all, I think that discussion that you have in this paragraph is very helpful and would improve the paper a lot. Currently, both sec 1.2, sec. 3 and sec. 4, are rather vague on this.
> > > >
> > > > Now that we have clarified more or less clarified "bushy", I have a question about distinction of boosting "features" and boosting "labels/classifiers". It sounds to me that these are almost the same since these labels/classifiers can be thought of as features too in boosting.
> > > >
> > > > So does that mean that main contribution of this work is to propose a new way of designing features/weak learners in boosting with particular emphasize on resnets perhaps? In my view giving a more precise formulation of these concepts can strengthen the paper quite a bit.
> > > >
> > > > 2, 3 Thanks for clarifications.

---

> > > > > ### Author Response · Authors · 2018-01-05
> > > > > **Clarify differences between boosting features and boosting labels**
> > > > >
> > > > > Thank you for following up. We appreciate your time.
> > > > >
> > > > > We would like to clarify the difference between boosting features and boosting labels. These two seem similar intuitively, however, there is no existing work that proves a boosting theory (guaranteed 0 training error) by boosting features. Moreover, the special structure that a ResNet has entails more complicated analysis: telescaping-sum boosting, which has never been introduced before in the existing literature.
> > > > >
> > > > > One of our main contributions, as the reviewer said, is to interpret ResNet using a “new” boosting theory. The weak learners used here are different from the traditional weak learners. In that sense, we have introduced a general new boosting framework that could be used by many applications other than ResNet.
> > > > >
> > > > > Again, we appreciate your evaluation and feedback. Thank you!

---

### Official Review · AnonReviewer1 · 2017-11-29
**The paper contains nice ideas and experimental results are promising, but has non-negligible mistakes in theoretical parts which degrade the contribution.**

**Rating:** 4
**Confidence:** 4

**Review:**

Summary:
This paper considers a learning method for the ResNet using the boosting framework. More precisely, the authors view the structure of the ResNet as a (weighted) sum of base networks (weak hypotheses) and apply the boosting framework. The merit of this approach is to decompose the learning of complex networks to that of small to large networks in a moderate way and it uses less computational costs. The experimental results are good. The authors also show training and generalization error bounds for the proposed approach.

Comments:
The idea of the paper is natural and interesting. Experimental results are somewhat impressive. However, I am afraid that theoretical results in the paper contain several mistakes and does not hold. The details are below.

I think the proof of Theorem 4.2 is wrong. More precisely, there are several possibly wrong arguments as follows:
- In the proof, \alpha_t+1 is chosen so as to minimize an upper bound of Z_t, while the actual algorithm is chosen to minimize Z_t. The minimizer of Z_t and that of an upper bound are different in general. So, the obtained upper bound does not hold for the training error of the actual algorithm.
- It is not a mistake, but, there is no explanation why the equality between (27) and (28) holds. Please add an explanation. Indeed, equation (21) matters.

Also, the statement of Theorem 4.2 looks somewhat cheating: The statement seems to say that it holds for any iteration T and the training error decays exponentially w.r.t. T. However, the parameter T is determined by the parameter gamma, so it is some particular iteration, which might be small and the bound could be large.

The generalization error bound Corollary 4.3 seems to be wrong, too. More precisely, Lemma 2 of Cortes et al. is OK, but the application of Lemma 2 is not. In particular, the proof does not take into account of the function \sigma. In other words, the proof considers the Rademacher complexity R_m(\calF_t), of the class \calF_t, but, acutually, I think it should consider R_m(\sigma(\calF_t)), where the class \sigma(\calF_t) consists of the composition of functions \sigma and f_t in \calF_t. Talagrand’s lemma (see, e.g., Mohri et al.’ s book: Foundation of Machine Learning) can be used to analyze the complexity of the composite class. But, the resulting bound would depend on the Lipschizness of \sigma in an exponential way.

The explanation of the generalization ability is not sufficient. While the latter weak hypotheses are complex enough and would have large edges, the complexity of the function class of weak hypotheses grows exponentially w.r.t. the iteration T, which should be mentioned.

As a summary, the paper contains nice ideas and experimental results are promising, but has non-negligible mistakes in theoretical parts which degrade the contribution of the paper.

Minor Comments:
-In Algorithm 1, \gamma_t is not defined when a while-loop starts. So, the condition of the while-loop cannot be checked.

---

> ### Author Response · Authors · 2017-12-11
> **The proofs are correct after double check**
>
> Thank you for your detailed comments.  Our proofs are correct---see below for specifics:
>
> 1. Proof of Theorem 4.2. Our proof is correct after double checking. The  \alpha_{t+1} is chosen to minimize Z_t in the algorithm. Therefore Z_t, achieved by choosing \alpha_{t+1} that minimizes Z_t, will be smaller than any other Z_t. In other words,  Z_t | {\alpha_t+1 = arg min {Z_t} } is less than or equal to Z_t | {\alpha_t+1 = anything other than arg min{Z_t} }. Therefore the upper bounds on Z_t in equation (31) and equation (78) hold.
>
> 2. Equation (21) is exactly the reason why the equality holds. We will add one line of explanation in the revised version.
> The statement of Theorem 4.2 is (unfortunately) split across pages, so please make sure to read the entire theorem. Theorem 4.2 clearly states that the statement is true only when weak learning condition is satisfied, not for all T.  The T required to achieve some error rate is certainly dependent on gamma as is specified in the complete theorem statement.
>
> 3. If Lemma 2 of Cortes et al. is right, then our proof will be right as well. Because the hypothesis class (equation (2) in Cortes et al. ) in Cortes et al. is the same as the hypothesis class  (equation (39) in our paper ) in our paper.  The results in Cortes et al. have to do with the relu function which is 1-Lipschitz activation function.
>
> 4. The generalization bound is stated explicitly in corollary 4.3. We suggest that small l_1 norm of weights help in terms of generalization.

---

### Official Review · AnonReviewer4 · 2018-01-02
**theoretical analysis is interesting, experiments are relatively weak**

**Rating:** 5
**Confidence:** 3

**Review:**

This paper formulates the deep ResNet as a boosting algorithm. Based on this formulation, the authors prove that the generalization error bound decays exponentially with respect to the number of residual blocks. Further, a greedy block-wise training procedure is proposed to optimized ResNet-like neural networks. The authors claim that this algorithm is more efficient than standard end-to-end backpropagation (e2eBP) algorithm in terms of time and memory consumption.
Overall, the paper is well organized and easy to follow. I find that using the boosting theory to analyze the ResNet architecture quite interesting. My concerns are mainly on the proposed BoostResNet algorithm.
1.	I don’t quite understand why the sequentially training procedure is more time efficient than e2eBP.  It is true that BoostResNet trains each block quite efficiently. However, there are T blocks need to be trained sequentially. In comparison, e2eBP updates *all* the blocks at each training iteration.
2.	The claim that BoostResNet is memory efficient may not hold in practice. I agree that the GPU memory consumption is much lower than in e2eBP. However, this only holds *under the assumption that the intermediate outputs of a previous block are stored to disk*. Unfortunately, this assumption is not practical for real problems: the intermediate outputs usually requires much more space of the original datasets. What makes thing worse, the widely used data augmentation techniques (horizontal flip, shift, etc.) would further make the space requirement hundreds of or even thousands of times larger.
3.	The results in Figure 2 seem quite surprising to me, as the ResNet architectures is supposed to be quite robust when the network goes deeper. Have you tried the convolutional ResNet structure used in their original paper?
4.	In Figure 3, how did you measure the number of gradient updates? In the original ResNet paper, the number of iterations required to train a model is 164(epochs)*50000(training samples)/128(batch-size)=6.4x10^5, which is far less than that showing in this figure.
5.	In Figure 3, it seems that the algorithms are not fully converged, and on CIFAR-10 the e2eBP outperforms BoostResNet eventually. Is there any explanations?

---

> ### Author Response · Authors · 2018-01-04
> **Clarifications on some misunderstandings**
>
> Thank you for your review. We would like to clarify the issues you pointed out as follows.
>
> 1.  The efficiency of BoostResNet is proved theoretically and justified empirically.  Theoretically,  the number of gradient updates required by BoostResNet is much smaller than e2eBP as discussed in Section 4.3. In practice, our experiments show significant improvement of computational efficiency as shown in figures 2 and 3.
>
> 2. During training, we don't consider a sample and an augmented sample as different. At each batch of training, samples are randomly augmented (cropped and horizontally flipped) before backpropagation - we don't precompute all possible augmentations and store it. This is consistent with standard training, where all possible augmentations are not stored on disk before training.
>
> 3. This is because the original ResNet paper uses convolutional modules, which enforce sparsity and have a regularizing effect. Figure two, as mentioned, is entirely fully-connected, as we said in the caption "on multilayer perceptron residual network". We compare BoostResNet with e2eBP on both residual convolutional networks and residual multilayer perceptron networks.
>
> 4. We are measuring the number of gradient updates, not the number of training iterations. The number of gradient updates increases by one every time any parameter in the network is updated.
>
> 5. We assure the readers that the results are fully converged. It is true that e2eBP eventually beats BoostResNet, however training using BoostResNet with small number of iterations and then fine tuning using e2eBP will be much more efficient than e2eBP alone. Because BoostResNet converges to some relatively good solution very fast. Due to the limited space, we move all the detailed discussions to Appendix H2.
>
> Again, we thank the reviewer for the review. We would like to emphasize that the contribution of the paper is multiple folds as discussed in section 1.1.  In particular, we hope to provide some training methods that have potential for training non-differentiable architectures (for example, tensor decomposition has been used to successfully train one layer neural network. Therefore we could potentially use BoostResNet where the training of one layer neural network is replaced  by tensor decomposition).

---

### Decision · Program_Chairs · 2018-01-29
**ICLR 2018 Conference Acceptance Decision**

**Decision:**

Reject

**Comment:**

All three reviewers felt that the paper was just below the acceptance threshold, with scores of 5,4,5. R1 felt there were problems in the proofs, but the authors rebuttal satisfactorily addressed this. R3 and the authors had an extended discussion with the authors, but did not revise their score from its initial value (5). R4 had concerns about the experimental evaluation, that wasn't fully addressed in the rebuttal. With no reviewers advocating acceptance, the paper will have to rejected unfortunately.